# Effect of Native Reservoir State and Oilfield Operations on Clay Mineral Surface Chemistry

**DOI:** 10.3390/molecules27051739

**Published:** 2022-03-07

**Authors:** Isah Mohammed, Dhafer Al Shehri, Mohamed Mahmoud, Muhammad Shahzad Kamal, Olalekan Alade, Muhammad Arif, Shirish Patil

**Affiliations:** 1Petroleum Engineering Department, College of Petroleum Engineering and Geosciences, King Fahd University of Petroleum and Minerals, Dhahran 31261, Saudi Arabia; g201806900@kfupm.edu.sa (I.M.); patil@kfupm.edu.sa (S.P.); 2Center for Integrative Petroleum Research (CIPR), College of Petroleum Engineering and Geosciences, King Fahd University of Petroleum and Minerals, Dhahran 31261, Saudi Arabia; shahzadmalik@kfupm.edu.sa (M.S.K.); olalekan.alade@kfupm.edu.sa (O.A.); 3Petroleum Engineering Department, Khalifa University, Abu Dhabi 127788, United Arab Emirates; muhammad.arif@ku.ac.ae

**Keywords:** clays, wettability alteration, low salinity water, adsorption, double layer effect

## Abstract

An understanding of clay mineral surface chemistry is becoming critical as deeper levels of control of reservoir rock wettability via fluid–solid interactions are sought. Reservoir rock is composed of many minerals that contact the crude oil and control the wetting state of the rock. Clay minerals are one of the minerals present in reservoir rock, with a high surface area and cation exchange capacity. This is a first-of-its-kind study that presents zeta potential measurements and insights into the surface charge development process of clay minerals (chlorite, illite, kaolinite, and montmorillonite) in a native reservoir environment. Presented in this study as well is the effect of fluid salinity, composition, and oilfield operations on clay mineral surface charge development. Experimental results show that the surface charge of clay minerals is controlled by electrostatic and electrophilic interactions as well as the electrical double layer. Results from this study showed that clay minerals are negatively charged in formation brines as well as in deionized water, except in the case of chlorite, which is positively charged in formation water. In addition, a negative surface charge results from oilfield operations, except for operations at a high alkaline pH range of 10–13. Furthermore, a reduction in the concentrations of Na, Mg, Ca, and bicarbonate ions does not reverse the surface charge of the clay minerals; however, an increase in sulfate ion concentration does. Established in this study as well, is a good correlation between the zeta potential value of the clay minerals and contact angle, as an increase in fluid salinity results in a reduction of the negative charge magnitude and an increase in contact angle from 63 to 102 degree in the case of chlorite. Lastly, findings from this study provide vital information that would enhance the understanding of the role of clay minerals in the improvement of oil recovery.

## 1. Introduction

The world today, even in the wake of renewable energies, is still largely driven by crude oil, and with the world’s population continuously increasing, efficient ways to produce more oil to meet demand is a necessity [1]. Crude oil is produced from reservoirs, either from sandstone or carbonate, with its production largely controlled by pressure, rock mineralogy, wettability state, and operational strategies. Reservoir rocks are predominantly made up of quartz or carbonate minerals with varying proportions of clays [2] and feldspar [3]. The most common group of individual clays found in the reservoir rocks are chlorite, illite, kaolinite, smectite, montmorillonite, or a mixture of clay layers [4]. These clays, when dispersed in the reservoir formation, predominantly control the pore space properties such as permeability and porosity of the rock. Furthermore, the high surface area and high cation exchange capacity controls the properties of mineral interactions, especially with the polar crude oil compounds.

The most polar compounds in crude oil are asphaltene and resins [5]. Asphaltenes are defined as a solubility class that is soluble in aromatics and insoluble in alkanes, and they have been reported to be responsible for the change in the reservoir rock wettability [5]. However, for asphaltenes to interact with surfaces, adsorption-prone surface conditions must exist, as asphaltenes have been reported to be predominantly negatively charged [6]. The influence of rock minerals, such as clays on asphaltene adsorption, is the subject of another ongoing study; however, of interest to this study is the surface chemistry of clay minerals to access the propensity of these minerals to provide an adsorption-prone surface condition. This is critical, as a previous study [4] shows that, because the reservoir rock is composed of many minerals, an understanding of the crude contacting minerals is critical to prevent polar compound adsorption on the mineral surfaces.

Considerable research in the literature exists on clay mineral interactions, with fluids and molecules resulting in their adsorption. Ebaga-ololo and Chon [7] reported the adsorption of linear and branched dodecyl alkyl sulfate (DAS) on a kaolinite surface using an adsorption experiment. The authors studied the adsorption isotherms of the pure and blended linear and branched DAS on a kaolinite surface, which was used to model the reservoir. The findings showed that the highest adsorption was observed with the 50–50 blend, and the adsorption behavior was best represented by the Langmuir isotherm. Fauziah et al. [8] reported the effect of CO_2_, brine, oil, and nitrogen on the wettability of montmorillonite, illite, and kaolinite at elevated pressures. The findings showed that in the presence of CO_2_, montmorillonite had the highest water contact angle followed by illite and then kaolinite. Furthermore, the adsorption of resins on the kaolinite edge surface has been reported [9]. Recently, Chen et al. [10] investigated acid and basic component interactions with kaolinite surfaces as a function of brine salinity using molecular dynamic (MD) simulations. They reported that the acid components were more responsive to salinity change and were responsible for kaolinite wetness. In addition, water molecules were observed to accumulate adjacent to the alumino layer compared to the silicate layer in a low saline environment, with the peak and position of molecules observed to be sensitive to the salinity level. Thus, a conclusion was reached that the adsorption density of species rather than layer thickness is controlled by salinity.

Chen et al. [11] reported the interaction of CO_2_ and N_2_ on K-illite surfaces using the Grand Canonical Monte Carlo simulation (GCMC). The results showed that the effect of static or dynamic potassium ions in the simulation is negligible and that sorption potential overlaps in the micropores. Similarly, Chong and Myshakin [12] reported the competitive adsorption of CO_2_ and CH_4_ on an illite surface using the GCMC simulation. The results showed preferential adsorption of CO_2_ over CH_4_ on illite surfaces and thus a prospective recovery strategy. Recently, Loganathan et al. [13] confirmed the preferential adsorption of CO_2_ on the basal surface compared to the CH_4_. However, the edges showed a stronger affinity for the CH_4_ over CO_2_, with the same density as the bulk phase around the edge surfaces.

Interaction of a functionalized silicon probe tip with a montmorillonite surface was investigated via first-principle calculations by Alvim and Miranda [14]. The authors observed that a tip functionalized with sulfonic acid and ethylene glycol were selective to the basal oxygens. Peng et al. [15] reported the adsorption of alkylamine cations on montmorillonite (001) surfaces using DFT calculations. The authors demonstrated the use of orbital energies to ascertain the nature of interactions. Furthermore, it was revealed that the interactions were by electrostatic and hydrogen bonding formation. Li et al. [16] revealed, in their study of amino acids adsorption on montmorillonite, that the interaction with montmorillonite had electrostatic interactions, cation exchange, and hydrophilic interactions as the governing mechanism. Ulian et al. [17] reported water molecule adsorption on a montmorillonite (001) surface using DFT calculations. Their results revealed favorable adsorption of a water molecule on the cation (Na^+^ and Ca^2+^) sites, with the Ca site being the most favored.

Even with the numerous research results reported in the literature on clay mineral interactions and chemistry, there is no report in the literature as to how the clay minerals behave in a native reservoir environment. More scare is how oilfield–pH inducing operations, as well as engineered water, affect the surface chemistry of clay minerals. To this effect, for the first time, the influence of the reservoir environment, as well as oilfield operations on the surface charge development of clays, is assessed. This is to provide critical insights into the clay mineral surface charge development process and how oilfield operations would affect the wetting state of the reservoir via surface charge modifications. Furthermore, a correlation between the observed zeta potential values and contact angle is reported.

## 2. Materials and Methods

The experimental measurements of surface charge of chlorite, illite, kaolinite, and montmorillonite are presented in this study to provide insight into the wettability alterations caused by these minerals. Reported in this study as well is the role of oilfield operations on the wetting state of these minerals and how the brine compositions affect their surface charge development.

### 2.1. Materials

The materials used in this study are as reported by Mohammed et al. [4]. Mineral particles used are of average particle size of 11.28, 3.16, 4.57, and 2.64 µm, for chlorite, illite, kaolinite, and montmorillonite, respectively, with all chemicals used of American Chemical Society (ACS) reagent grade. Synthetic Arabian Gulf seawater (SW) and formation water (FW) with compositions as shown in Table 1 were used. Salts used to prepare the brine solutions included sodium chloride (NaCl), sodium sulfate (Na_2_SO_4_), sodium bicarbonate (NaCO_3_), magnesium chloride (MgCl_2_), and calcium chloride (CaCl_2_) from Sigma Aldrich (Saint Louis, MO, USA).

### 2.2. Zeta Potential Sample Preparation

Clay mineral surface charge development in different reservoir fluid conditions is presented in this study. The particles were conditioned in different fluids (deionized water (DI), formation water (FW), seawater (SW), and ion engineered water) of varying composition and salinity to mimic oilfield operations and a realistic reservoir environment. Throughout this study, 10 mg of clay sample particles was conditioned in 10 mL of fluid, with the mixture sonicated for 2 h to ensure maximum interactions. Thereafter, the mixture was allowed to stand undisturbed for 24 h before measurements were conducted. For the ion concentration studies, the different proportions of FW and SW were mixed to achieve the desired ionic composition. Direct dilution with DI water was not used in this study, as it does not reflect operations carried out in the field; thus, seawater was used for dilution and ion modification to achieve the target ion concentrations.

### 2.3. Zeta Potential (ζ) Measurement

Zeta potential measurement was conducted using the Anton Paar Litesizer 500 (Graz, Austria) with a size range of 3.8 nm to about 100 µm. Omega cuvette made of polycarbonate, with a volume of 350 µL was used. Before every zeta potential measurement, sample mixtures were centrifuged using Hermle Z326K centrifuge (Gosheim, Germany) equipped with a refrigeration system. Samples were centrifuged at 2000 rpm and 23 °C for 2 min before the measurement sample was taken from the clear supernatant. All reported measurements in this study are within ±2 mV standard deviation.

### 2.4. Contact Angle Measurement

The sample mineral (chlorite and illite) wettability was evaluated using contact angle measurement with the minerals conditioned in the respective fluids (DI, FW, and SW) 24 h before the measurements. The contact angle was measured between the mineral chip/fluid interface (Figure 1) in the presence of crude oil, whose composition is shown in Table 2. The measurement was conducted at 25 °C and atmospheric pressure using the Attension theta optical tensiometer (Phoenix, AZ, USA).

## 3. Result and Discussions

### 3.1. Reservoir Environment

Zeta potential (ζ) measurement of the clay particles in deionized (DI) water, formation water (FW), seawater (SW) as well as mixtures with different ion concentrations was conducted with standard deviation within ±2 mV. As an initial step to establish a benchmark for comparison, the surface charge of the clay particles conditioned in DI, FW, SW was measured to mimic the different environments possible in the oil reservoir. The pH values of the freshly prepared synthetic brine solutions, as well as the DI water used in this study, were 6.33, 5.94, and 8.06 for DI, FW, and SW, respectively (Figure 2). This serves as a benchmark for pH changes reported in this study and reveals the alkaline nature of seawater used for water injection. The result of each mineral conditioned in the different fluids for 24 h. is discussed in the subsequent sections.

#### 3.1.1. Chlorite

The measured ζ potential values of chlorite minerals in DI water, FW, and SW are shown in Figure 3. From this figure, chlorite mineral is negatively charged in DI water and SW, whereas it is positively charged in FW. To understand the observed surface charge of chlorite depicted in Figure 3, an understanding of its structure is critical. Chlorite is a phyllosilicate mineral with a 2:1 sandwich structure of tetrahedral–octahedral–tetrahedra (t–o–t) also commonly referred to as the talc layer. Unlike most clays, the space between the talc layer is filled with cation (Mg^2+^, Fe^3+^) (OH)_6_ units. Thus, given that the exposed surface is not known, either of these two surfaces ((Mg, Fe)_3_(OH)_6_ and (Si, Al)_4_O_10_(OH)_2_) could be responsible for the observed surface charge development. However, from the reduction in pH value of the mixture from an initial pH value of 6.3 to 5.72, (Si, Al)_4_O_10_(OH)_2_ is probably the exposed surface, as the chlorite behavior is close to the quartz mineral, which is under study in a separate ongoing research. Thus, the behavior reported here is attributed to (Si, Al)_4_O_10_(OH)_2_. Also, with the reduction of the pH value, from 6.33 to 5.72, it can be inferred that the interaction responsible for the observed surface charge is between the OH_water_ ion and the Si_chlorite_, resulting in excess H_water_ ions and exposed O_chlorite_ ions. This is responsible for the slight reduction in the pH.

For chlorite particles in FW, a positive surface charge is observed with no pH change. This means that H^+^ and OH^−^ are not potential-determining ions, and the observed positive charge is due to the presence of other ions in the system. The positive surface charge can be attributed to the cations present in the system; however, because of the different hydration stabilities of the cations, the observed positive surface charge is due to the Na atom rather than the Mg or Ca atoms, as their hydration stability would not allow them to approach the surface. In addition, the mode of cation interaction with the surface is via adsorption, as earlier reported by Mohammed et al. [4], in the same pH range. In the case of SW, no pH change is observed, with the surface charge of chlorite being negative. This implies that the surface charge is not controlled by H^+^/OH^−^ ions of water, but by other constituent ions in the SW. With the surface being negatively charged, the anions in the system are responsible for the observed surface charge, with the order of dominance being Cl^−^ > SO_4_^2−^ > HCO_3_^−^.

#### 3.1.2. Illite

The surface charge of illite particles in DI water, FW, and SW are shown in Figure 4. There is no observable change in the pH across all mediums, inferring that H^+^ and OH^−^ are not potential-determining ions in the case of illite; thus, the observed charges are due to the fluid composition. In the case of DI water, the illite mineral behavior is similar to that of other phyllosilicate minerals, with the dominant interaction via H atoms (H^+^ and OH^−^); however, no pH change exists in the case of chlorite. Furthermore, the negative surface charge observed in the cases of FW and SW shows the dominance of the water content of illite in controlling the ion interactions with the surface; however, due to the presence of the ions in both FW and SW, a reduction in the negative charge magnitude is observed. An all-negative surface charge implies that the illite mineral in both FW and SW compared to chlorite results in surfaces that are less prone to polar crude oil adsorption. Although chlorite positive surface charge in FW is small, it can easily be reversed using ion-engineered fluids or chelating agents.

#### 3.1.3. Kaolinite

The experimentally measured zeta potential values of the kaolinite particles in DI water, FW, and SW are shown in Figure 5. Comparison of the surface charge behavior to that of the illite particles (Figure 4) and those of kaolinite (Figure 5) shows similar behavior, except for the pH change observed in the case of kaolinite in DI water. Compared to the case of illite, where H^+^ and OH^−^ are not potential-determining ions, kaolinite in DI water pH change shows that there exist interactions between the OH_water_ and the surface (hydrogen bonding). Furthermore, like the case of illite, the kaolinite surface charge behavior in all fluids (DI water, FW, and SW) mitigates the adsorption of polar crude compounds such as asphaltenes and resins. However, it contains less water in its structure compared to illite.

#### 3.1.4. Montmorillonite

The behavior of montmorillonite particles in DI water, FW, and SW are shown in Figure 6, and as discussed earlier for other minerals, their behavior is like those of kaolinite. Furthermore, a slight pH change in DI water is observed such as in the case of kaolinite. This can be attributed to the similar structures of both minerals. Montmorillonite and kaolinite are similar in structure, with the only difference being that montmorillonite has two silica tetrahedral sheets, with kaolinite having one silica tetrahedral sheet. In addition, montmorillonite absorbs more water than kaolinite, with illite having intermediate water absorption.

### 3.2. Effect of Oilfield Operations

Different oilfield operations are implemented during primary production (FW) and after seawater, (SW) injection for pressure maintenance and flooding purposes. These operations, which include acidizing, stimulations, surfactant, alkaline flooding, etc., induce pH change around the wellbore, affecting the surface chemistry of the minerals within these environments. These effects have dire consequences, as they change the surface condition of the minerals, thus altering the nature of interaction with reservoir fluids. The effect of these operations was mimicked by the pH change caused by it. Thus, the surface charge behavior of clays due to oilfield induced-pH is reported.

#### 3.2.1. Chlorite

The effect of oilfield operations on the chlorite mineral surface charge is shown in Figure 7. The effects were studied in two stages. The first is the effect of oilfield operations during the primary production stage (FW), where the minerals are still immersed in their native environment. The second stage is during the secondary production stage (SW) when seawater injection has been implemented. This approach was adopted to have a comprehensive understanding of the effect in both environments. In the primary production stage (FW), it can be observed that around the acidic (1–3) pH range, the minerals were negatively charged, although the magnitude of the surface charge is small; however, toward a neutral pH (4–5), a charge reversal is observed. This agrees well with the earlier observation of the mineral surface charge in Figure 3. At neutral pH values (6–8), a negatively charged surface is observed, with a charge increase from pH 7–8 before a charge reversal. At alkaline pH values of 11–13, a positive surface charge is observed, with an increase in magnitude with increasing pH, which is noted from the trends in the surface charge increase and charge reversals across the pH values. This is attributed to the existence of an electrical double layer phenomenon. For example, in the alkaline pH region, an increase in pH from 11 to 13 results in increased OH^−^ density. Thus, one would expect that the surface should be negatively charged; however, an increased positive surface charge magnitude is observed due to the compression of the positive ions in the stern layer on the particle surface, whereas the OH^−^ are in the diffuse layer. A similar double layer compression effect is observed in the acidic pH range (1–3) as the H^+^ density increases with pH decrease.

In the case of the SW, which represents oilfield operations implemented after seawater injection, an all-negative surface charge is observed, except for a few pH values (1 and 5), where charge reversal was observed. This is because the factors responsible for the negative surface charge, as explained earlier, are dominant in play, although pH modification has been implemented. Furthermore, the electrical double layer effect (reduction in the negative charge magnitude with increased OH^−^ density at extreme alkaline pH) can be observed in the case of SW, however with a higher magnitude of surface charge than in the case of FW. Furthermore, the chlorite mineral can largely be said to be negatively charged across pH, with surfaces that can mitigate a crude oil polar compound based on its surface charge behavior, although an interplay of other factors such as extended exposure to acidic or alkaline pH might cause charge reversal.

#### 3.2.2. Illite

The effects of oilfield operation on the surface charge behavior of illite particles are depicted in Figure 8. In the case of FW, which represents the illite particles in the native reservoir environment, a negative surface charge is observed from the acid to neutral pH range (1–8), which agrees well with our earlier measurements shown in Figure 4. However, at pH 9 and 11–12, a charge reversal is observed, resulting in a positive surface charge. This charge reversal can be attributed to the collapse of the double layer at the alkaline pH region. Furthermore, across the pH values of 1–8, the effect of double-layer compression is apparent; however, a collapse only resulted in the alkaline pH region. This means that, in the native reservoir state, the illite mineral does not pose a threat of serving as a precursor for polar crude oil compounds and is thus not responsible for any observed wettability alterations caused by those compounds. Even more interesting is its behavior in the secondary production state (seawater injection implemented) across all pH. In this case (SW), an all-negative surface charge is observed, with a seemingly stable value at a pH range of 5–11. Although the effect of double-layer compression is observed at extreme pH values (12–13 and 1–3), it does not result in charge reversal. This supports the opinion that the illite particle does not partake in reservoir rock wettability alteration, as their surfaces are mostly waterwet across all pH, especially in the secondary production stage.

#### 3.2.3. Kaolinite

The effect of oilfield operations on kaolinite particle surface charge is shown in Figure 9, with its behavior like that of the chlorite particles. This is not surprising because a kaolinite-to-chlorite conversion has been recently reported in the literature [20]. In the case of FW, an all-negative surface charge across the pH values is observed, except for a few pH values (9, 12, and 13), where a positive surface charge is recorded. Like the case of chlorite, the effect of double-layer compression is pronounced at pH values of 1–8, with the charge reversal occurring at extreme alkaline pH values. Conversely, in the case of SW, an all-negative surface charge is observed with the dominant effect being the double-layer compression. Like other minerals (chlorite and illite), the surface charge condition of the kaolinite particles does not provide an adsorption-prone surface for polar crude oil compound adsorption. Thus, they are not responsible for the wettability alterations in rocks; however, their large surface areas may provide enough surface for the deposition of polar crude oil compounds.

#### 3.2.4. Montmorillonite

The effects of oilfield operations on the surface charge development of montmorillonite are shown in Figure 10. In both cases of FW and SW, which represent the effect of oilfield operations on montmorillonite particles’ surface charge, an all-negative surface charge is observed, except for the alkaline pH region, where charge reversal is recorded. From both trends, the effect of double-layer compression (pH 1–11) and thereafter double-layer collapse (pH 12–13) is worth noting. From these observations, like other particles discussed, clay minerals do not potentially present threats of polar compound adsorption due to pH-induced oilfield operations. In addition, the injection of SW results in a higher magnitude of negative surface charge, thus enhancing the ability of the surfaces to mitigate polar crude oil compounds adsorption.

### 3.3. Effects of Ion Engineering

The injection of ion-engineered water, or low salinity water as it is popularly referred to, has gained acceptance in the industry. Several works of literature [21,22,23,24,25,26,27,28] exist on oil recovery improvement due to ion-tuned water injection, with several mechanisms identified to be responsible for the observed recovery. This section examines the effect of ion-tuned water on the surface charge development of the particles under study. To this effect, particles were conditioned in ion-tuned water with their surface charge determined.

#### 3.3.1. Chlorite

The effects of ion-tuned water on the surface charge development of chlorite particles are shown in Figure 11 and Figure 12. To examine which ions control the surface charge development of chlorite particles, the concentration of the specific ions (cations) was reduced by 25%, 50%, and 75%. However, in the case of sulfate, an increase in its concentration was studied instead, as the seawater used as an injection fluid has more sulfate ions than the formation brine. Figure 11 shows the case of Na, Mg, Ca, bicarbonate, and chloride ion reduction on the chlorite particles’ surface charge. As can be observed from the trends, reduction in the concentration of the cations (Na, Ca, and Mg) results in increased negative surface charge, with the most increase observed in the cases of Na and Mg ions. Conversely, for a reduction in the case of anions (Chloride and bicarbonate), only bicarbonate ions show increased surface charge with a reduction in its concentration, which infers that it is not a dominant ion in the surface charge determination compared to the chloride ion. Furthermore, reduction in the bicarbonate ion concentration reduces the surface interaction competition for the chloride ion, thus giving room for the surface to interact more with the chloride ions and thus an observed increase in the surface charge magnitude. For the chloride ions, a reduction of more than 50% does not significantly affect the surface charge, as an almost constant value of the surface charge is recorded. In addition, the observed decrease in surface charge magnitude with concentration reduction from 25–50% shows that the chloride ion is a potential-determining ion in the case of chlorite particles.

In the case of the sulfate ion concentration increase shown in Figure 12, a 25% increase in the sulfate ion concentration results in a positive surface charge owing to the double-layer collapse. In addition, a continuous increase in concentration reversed the charge to negative, and above a 75% concentration increase, the dominance of the double-layer collapse became apparent, as, after this concentration, an alternating surface charge between positive and negative was observed. This indicates that, in the implementation of ion-tuned water injection with increased concentration of sulfate ion, an optimum concentration must be determined *apriori*.

#### 3.3.2. Illite

The effects of ion-engineered water on illite particle surface charge are shown in Figure 13 and Figure 14. Figure 13 shows the effects of a reduction in the Na, Ca, Mg, bicarbonate, and chloride ions concentration on the illite particle surface charge. As depicted in this figure, reduction in both cation and anion concentration has a significant impact on the surface charge magnitude. Thus, ion-engineered water can be significant in controlling the behavior of illite particles. Conversely, an increased concentration of the sulfate ion results in a more negatively charged surface, with the maximum negative value recorded at a 50% sulfate ion increase. The reduction of the surface charge magnitude and subsequent charge reversal (Figure 14) can be attributed to the dominance of the electrical double layer phenomena. Thus, like chlorite particles, optimum sulfate ion concentration is needed to achieve maximum negatively charged surface conditions.

#### 3.3.3. Kaolinite

The effects of ion-tuned water on kaolinite particle surface charge are shown in Figure 15 and Figure 16. The reduction of Na and Mg ion concentrations shows increased negative surface charge, whereas in the case of the Ca ion beyond 25% ion concentration reduction, no significant effect is observed. Thus, it can be inferred that the order of effect of the cations is Na > Mg > Ca. In the case of anions (chloride), the most significant effect is observed at 75% ion reduction, which also means a reduction in the Na, Ca, and Mg ions since they contribute to the chloride ion concentration in the system. The case of bicarbonate is slightly different, as a 25% reduction results in a positive surface charge; however, further reduction results in a negative surface charge, with not much difference between the magnitude at 50% and 75% ion reduction. Conversely, an increase in sulfate ion concentration (Figure 16) up to 50% does not show any significant change in surface charge magnitude, with charge reversal observed with a further increase in sulfate ion concentrations. Moreover, the subsequent increase in the sulfate ion concentration shows the dominance of the double-layer collapse phenomena. Thus, it can be concluded that the sulfate ion has less effect on the kaolinite particle surface charge.

#### 3.3.4. Montmorillonite

The effects of ion-tuned water injection on montmorillonite particle surface charge are shown in Figure 17 and Figure 18. From Figure 17, it can be inferred that the optimum ion reduction percentage for Na, Ca and Mg ion is 25%, which shows the highest negatively charged surface as a further reduction resulting in reduced surface charge magnitude. Conversely, a slight increase in the magnitude of the surface charge is observed in the case of the bicarbonate, with the most significant effect observed to be from the chloride reduction. Furthermore, as in the case of other minerals, the effect of the electrical double layer is observed in the case of sulfate ion increase (Figure 18).

### 3.4. Effect of Salinity Gradient

In all reservoirs where water injection is implemented, there exists a salinity gradient from the injection wells to the production wells. This creates different environments as the injection water progresses through the reservoir. In the case of chlorite mineral (Figure 19), the surface charge at the injection well and near the injection well is negatively charged; however, as the fluid attains a 50:50 mix ratio of the formation water, the surface charge becomes positive. This does mean that to establish an all-negative surface charge, injected seawater must have broken through the producer well. Directly before the reversal of the surface charge, an almost-zero surface charge is observed at the 50:50 mix ratio. In the case of illite (Figure 20), at the injector well location and near the injector well, a negative surface charge is observed even at a 50:50 mix ratio as compared to the chlorite particles, where charge reversal occurred at a 50:50 mix ratio. The charge reversal however occurred at 75% FW. Contrary to the observation of charge reversal in the case of chlorite and illite particles, kaolinite (Figure 21) and montmorillonite (Figure 22) show an all-negative surface charge across the salinity gradient. Thus, the effect of salinity gradient on clay minerals creates inhibitive surface conditions to polar crude oil compound adsorption.

### 3.5. Contact Angle Correlation

Contact angle indicates the state of wetness of a rock, with its values ranging between 0 and 180. A contact angle of zero indicates complete water wetness, and a value of 180 indicates oil wetness. Correlation between zeta potential values and contact angle has been reported in the literature [29]; however, caution must be exercised when attempting to establish a correlation. This is because the zeta potential value as a wettability alteration indicator is dependent on what the wetting phase is, and the explanations of the observed changes must acknowledge the reference state before alterations. In the study by El-Din Mahmoud [29], the effect of chlorite clay minerals on recovery was evaluated. A similar approach to that of El-Din Mahmoud [29] is adopted here, but this time, pure mineral samples (chlorite and illite) were used instead of core samples with varying fractions of chlorite mineral. A good correlation can be established between the zeta potential mineral surface charge with the contact angle measured in different fluids. From Figure 23, it is observed that increasing fluid salinity results in a positive surface charge and an increased contact angle. Thus, increased salinity corresponds to an increase in the contact angle, inferring less water wetness. Similarly, evaluation of the illite sample contact angle in different fluids (Figure 24) showed that the zeta potential value decreases with increased fluid salinity, whereas the contact angle increases. This thus again supports earlier explanations of increased water wetness with reduced fluid salinity. From the discussed results thus far, the negative surface charge of the clay minerals can be inferred to depict a water-wet condition of the surfaces. However, the water wetness can be improved by using chelating agents, which sequester the cations from the mineral’s surfaces.

## 4. Conclusions

This study experimentally measured the zeta potential of chlorite, illite, kaolinite, and montmorillonite particles in different fluids of varying composition and salinity and are reported with the following conclusions:1.Clay minerals, except for in the case of chlorite in the native reservoir environment, do not provide an adsorption-prone surface to polar crude oil components due to their surface charge.2.Oilfield operations do not impact the nature of the clay mineral surface charge; however, they affect the magnitude of the surface charge. Moreover, significant effects of the oilfield operations are dominant in the alkaline pH regions.3.The dominant mechanisms responsible for clay mineral surface charge development are the electrical double-layer effect and ion adsorption.4.Ion-engineered water has a significant effect on the clay mineral surface charge, especially in the case of sulfate ions.5.The salinity gradient across the reservoir results in different clay mineral surface charges across the reservoir.6.Increased fluid salinity results in an increase in the contact angle, which consequently leads to a less water-wet surface condition.

## Figures and Tables

**Figure 1 molecules-27-01739-f001:**
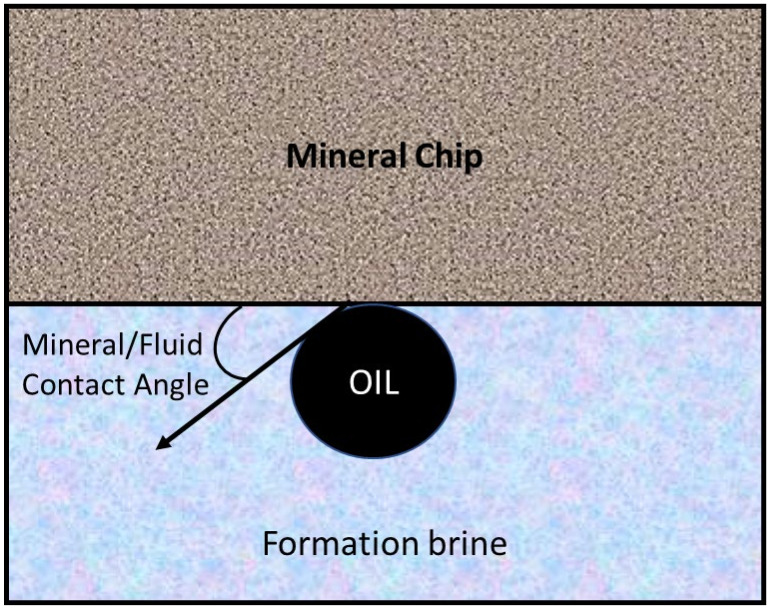
Contact angle measurement.

**Figure 2 molecules-27-01739-f002:**
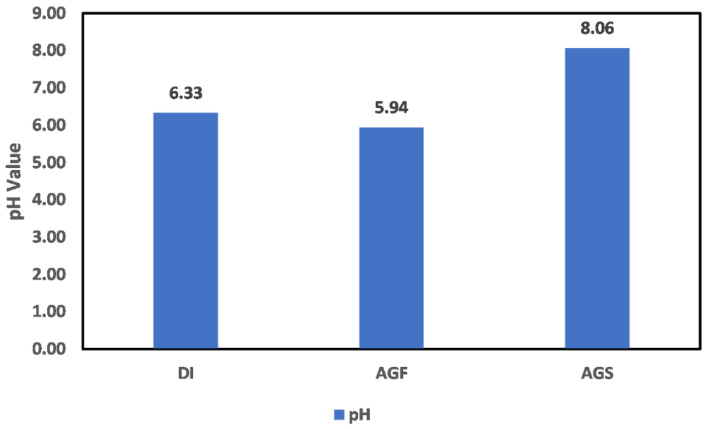
pH values of deionized water (DI), synthetic Arabian Gulf formation water (AGF), and Arabian Gulf seawater (AGS).

**Figure 3 molecules-27-01739-f003:**
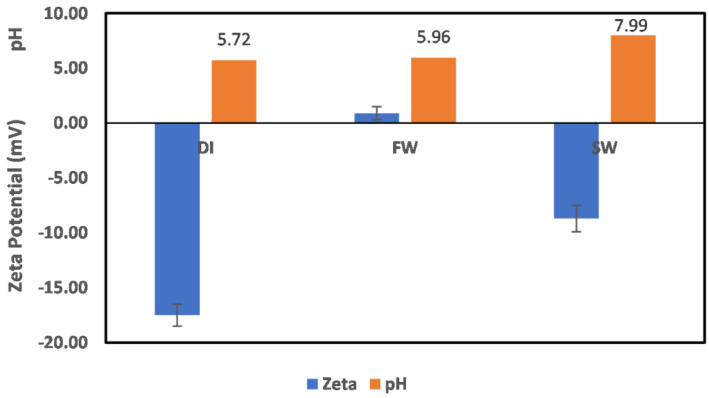
Chlorite particle surface charge in DI water, FW, and SW. Data label depicting the exact pH values is shown on the plots. This applies to all subsequent figures with pH values represented alongside the observed zeta potential values.

**Figure 4 molecules-27-01739-f004:**
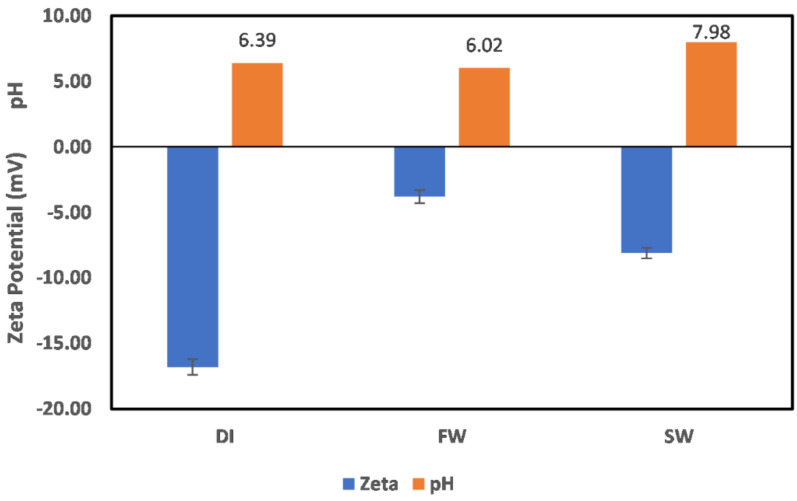
Illite particle surface charge in DI water, FW, and SW.

**Figure 5 molecules-27-01739-f005:**
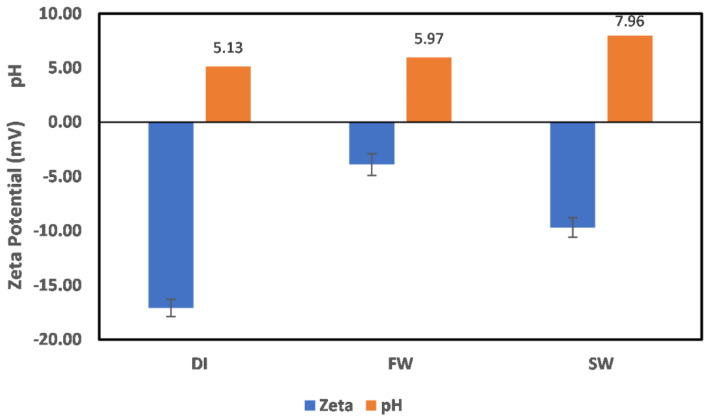
Kaolinite particle surface charge in DI water, FW, and SW.

**Figure 6 molecules-27-01739-f006:**
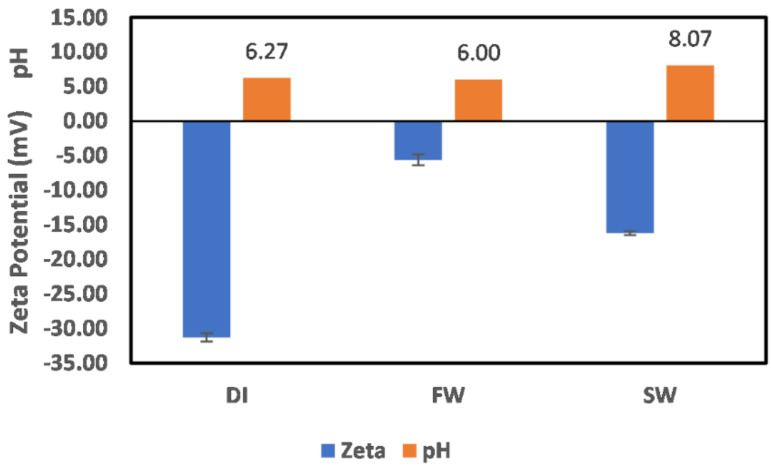
Montmorillonite particle surface charge in DI water, FW, and SW.

**Figure 7 molecules-27-01739-f007:**
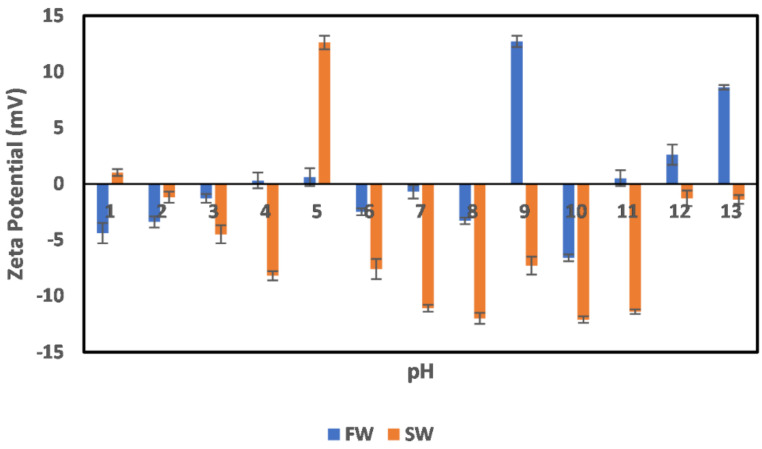
Effects of oilfield operations on chlorite mineral surface charge.

**Figure 8 molecules-27-01739-f008:**
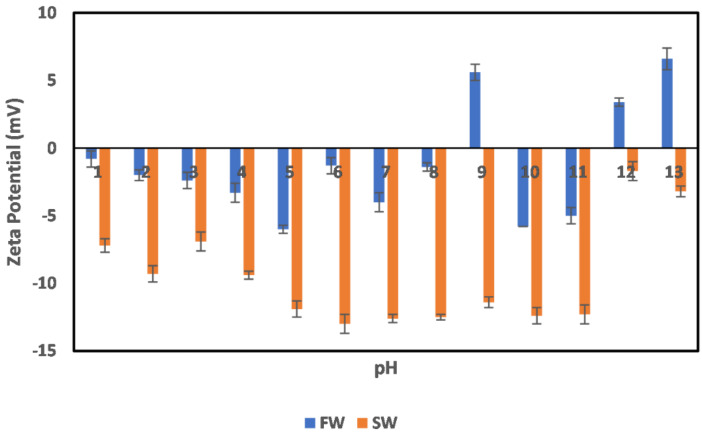
Effects of oilfield operations on illite mineral surface charge.

**Figure 9 molecules-27-01739-f009:**
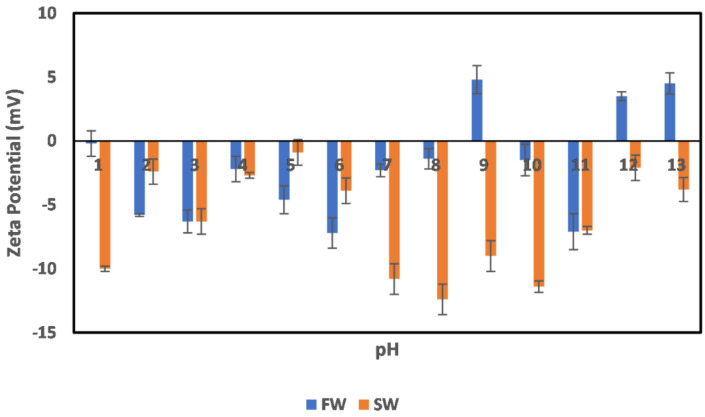
Effects of oilfield operations on kaolinite mineral surface charge.

**Figure 10 molecules-27-01739-f010:**
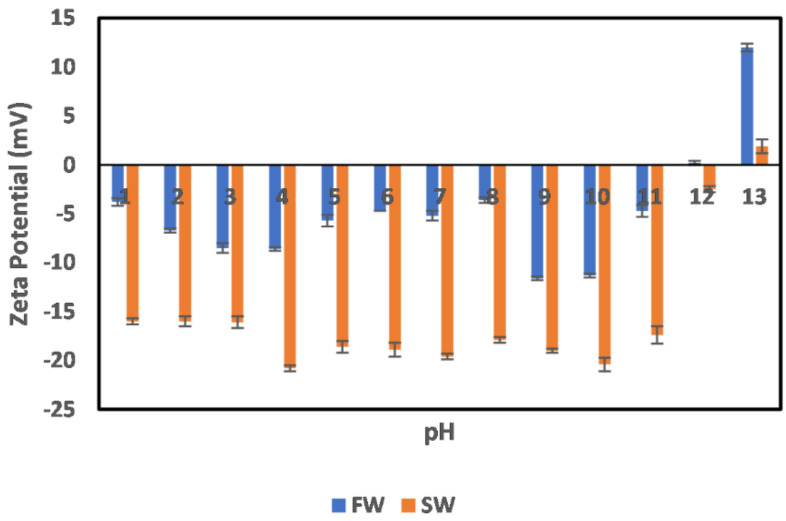
Effects of oilfield operations on montmorillonite mineral surface charge.

**Figure 11 molecules-27-01739-f011:**
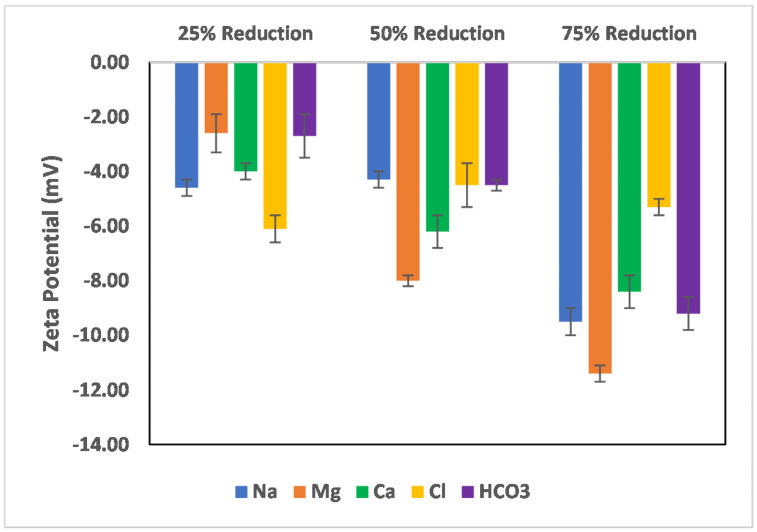
Effects of Na, Mg, Ca, chloride, and bicarbonate ion concentration reduction on chlorite particle surface charge.

**Figure 12 molecules-27-01739-f012:**
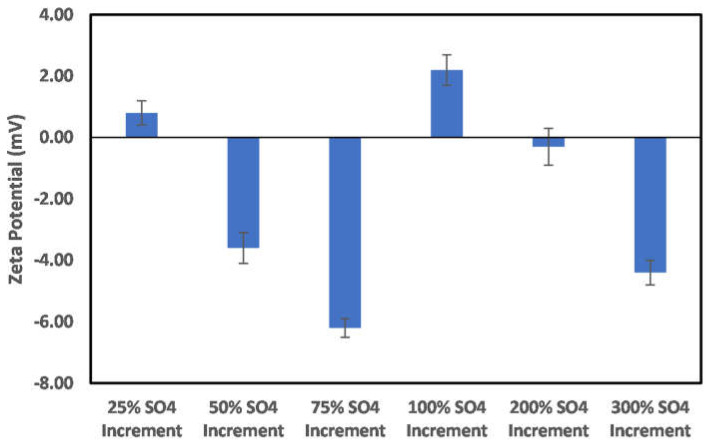
Effects of increase in sulfate ion concentration on chlorite particle surface charge.

**Figure 13 molecules-27-01739-f013:**
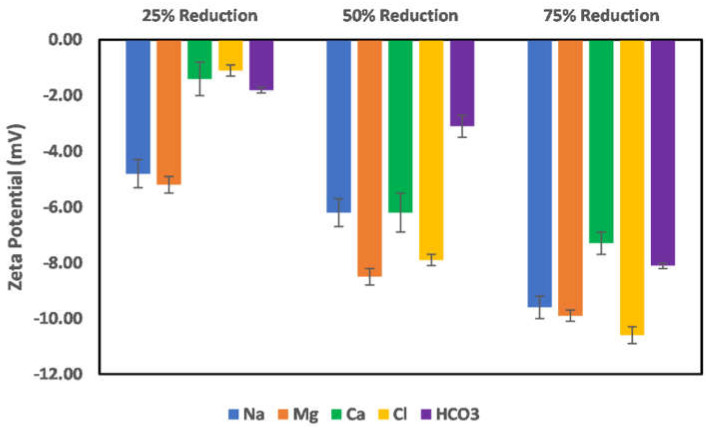
Effects of Na, Mg, Ca, chloride, and bicarbonate ion concentration reduction on illite particle surface charge.

**Figure 14 molecules-27-01739-f014:**
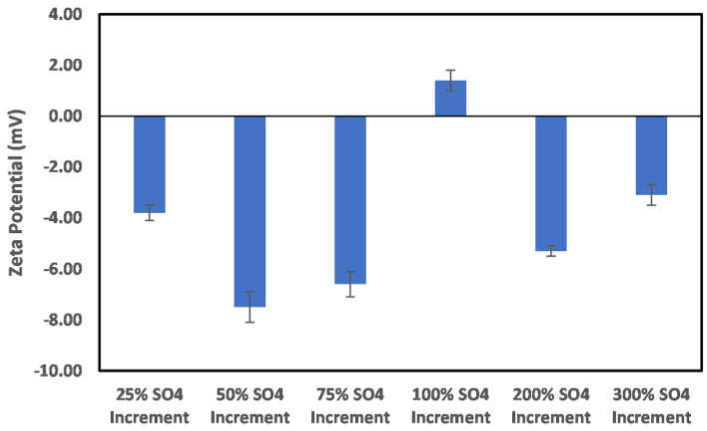
Effects of increase in sulfate ion concentration on illite particle surface charge.

**Figure 15 molecules-27-01739-f015:**
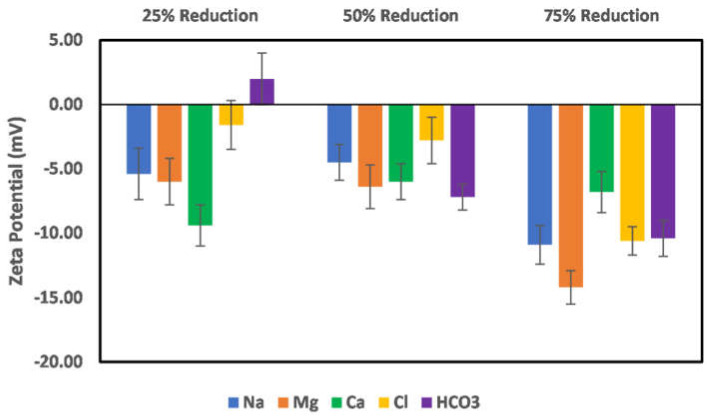
Effects of Na, Mg, Ca, chloride, and bicarbonate ion concentration reduction on kaolinite particle surface charge.

**Figure 16 molecules-27-01739-f016:**
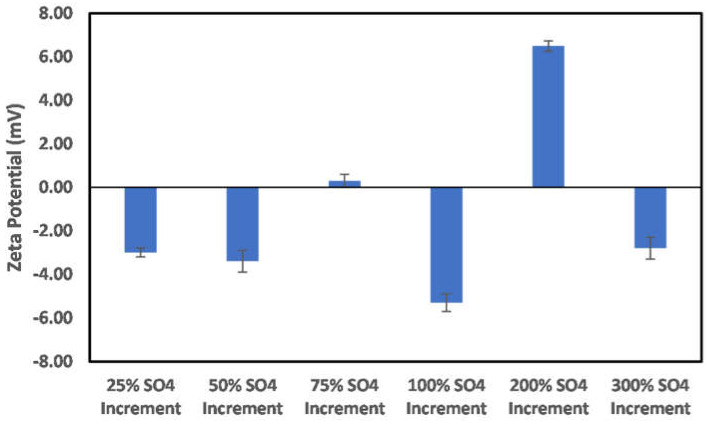
Effects of increase in sulfate ion concentration on kaolinite particle surface charge.

**Figure 17 molecules-27-01739-f017:**
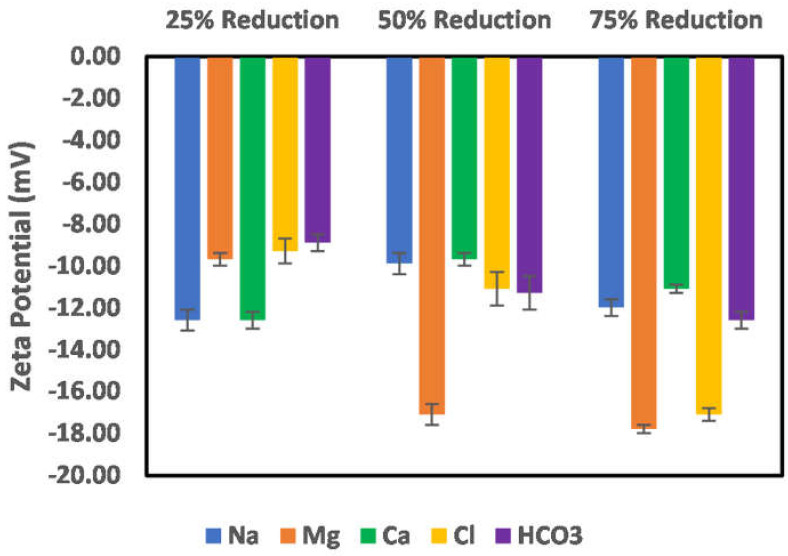
Effects of Na, Mg, Ca, chloride, and bicarbonate ion concentration reduction on montmorillonite particle surface charge.

**Figure 18 molecules-27-01739-f018:**
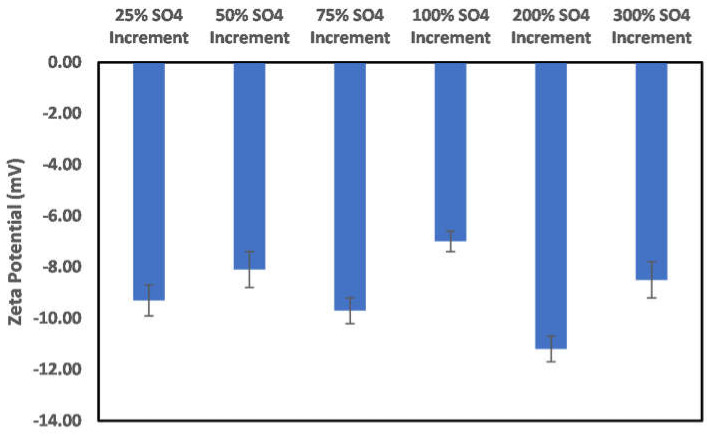
Effects of increase in sulfate ion concentration on montmorillonite particle surface charge.

**Figure 19 molecules-27-01739-f019:**
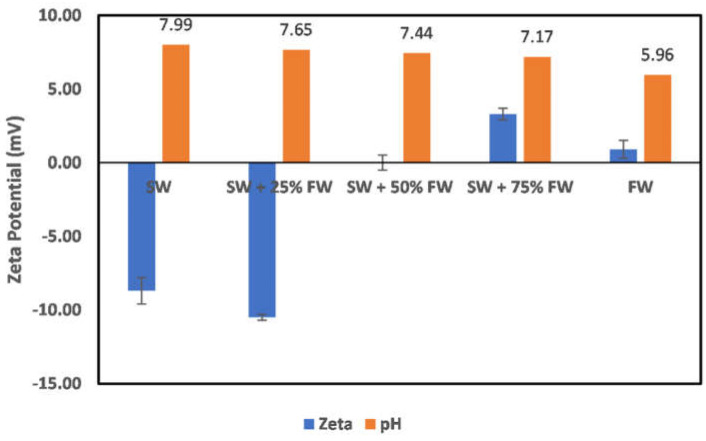
Effects of salinity gradient on chlorite particle surface charge.

**Figure 20 molecules-27-01739-f020:**
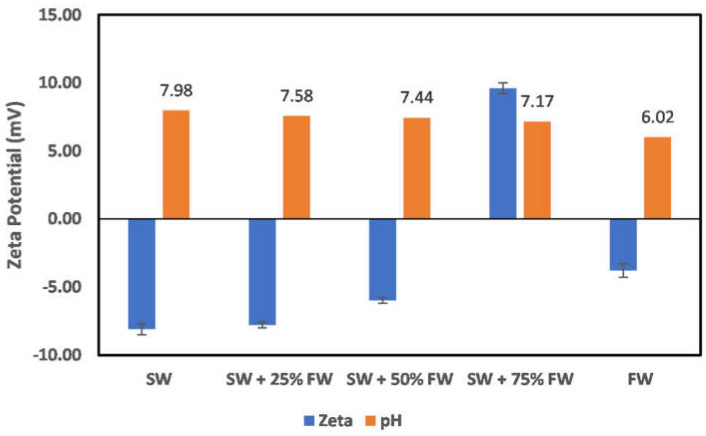
Effects of salinity gradient on illite particle surface charge.

**Figure 21 molecules-27-01739-f021:**
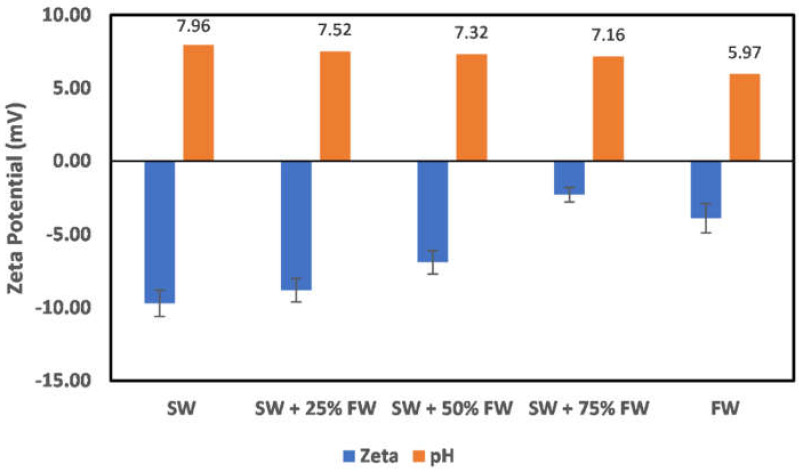
Effects of salinity gradient on kaolinite particle surface charge.

**Figure 22 molecules-27-01739-f022:**
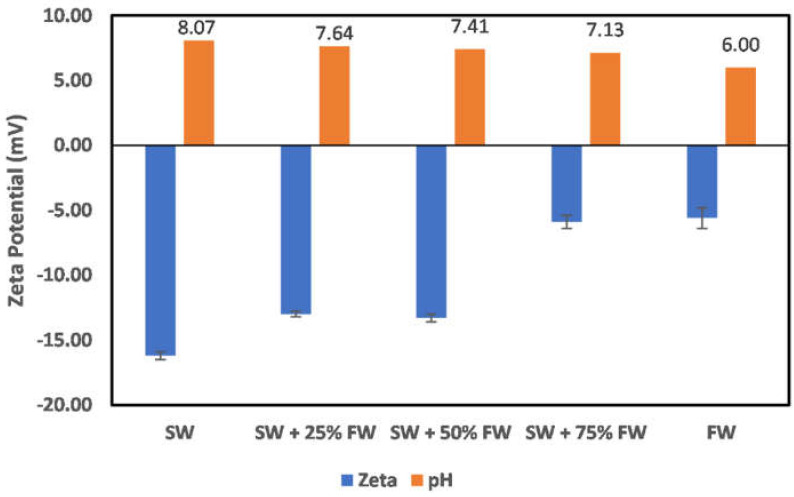
Effects of salinity gradient on montmorillonite particle surface charge.

**Figure 23 molecules-27-01739-f023:**
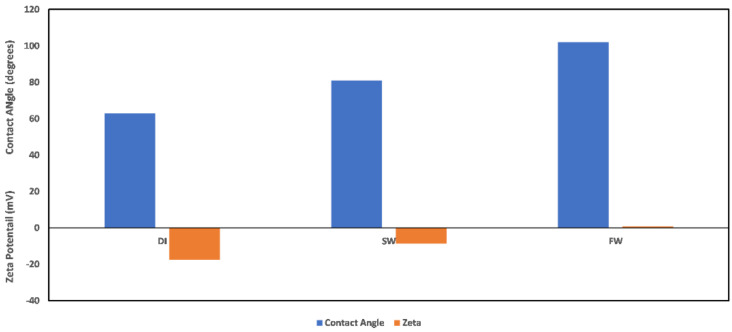
Contact angle and zeta potential correlation for chlorite sample.

**Figure 24 molecules-27-01739-f024:**
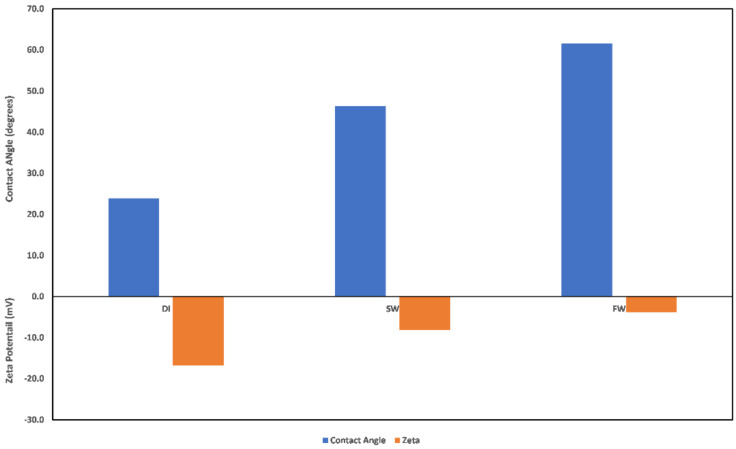
Contact angle and zeta potential correlation for illite sample.

**Table 1 molecules-27-01739-t001:** Ionic composition of formation water and Arabian seawater [18].

Ions	Formation Water Ions (ppm)	Seawater Ions (ppm)
Na^+^	59,491	18,300
Ca2^+^	19,040	650
Mg^2+^	2439	2110
SO_4_^2−^	350	4290
Cl^−^	132,060	32,200
HCO_3_^−^	354	120
TDS	213,734	57,670

**Table 2 molecules-27-01739-t002:** Physical parameters of crude oil used [19].

Physical Characterization/Mass Percentage	Value
API gravity at 15 °C	32.49
API specific gravity at 15 °C	0.863
Density (g/cm3) at 15 °C	0.8620
Viscosity (mPa·s) at 15 °C	10.9
SARA fractions:	
Saturates (wt.%)	36.2
Aromatic (wt.%)	50
Resins (wt.%)	11
Asphaltenes (wt.%)	2.8

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
