# Peer review of "Effect of Native Reservoir State and Oilfield Operations on Clay Mineral Surface Chemistry"

_molecules, 2022, doi:10.3390/molecules27051739_

Round 1

Reviewer 1 Report

Manuscript number: Molecules-1562464

Title: “Effect of Native Reservoir State and Oilfield Operations on 2 Clay Mineral Surface Chemistry”

Authors: Mohammed et al.

The results in the abstract section appears to be more qualitative than quantitative. Specific observations in the form of experimental results should be mentioned in the abstract.

Detailed description on clay structure (e.g., Page 2, Line 51-58) is out of the context.

Overall, the introduction section should substantially be shortened, instead of detail description, that seem more like review. The aim and scope, novelty statement may be provided at the end of the introduction section.

Please check line 122.

Although reported previously, the short description on the clay/mineral particles would be helpful for the general readers.

Geographical location of the Synthetic Arabian Sea Water may be provided.

In a number of places the statement appears “(Error! Reference source not found.)” Please check.

Cell type for zeta potential measurement should be mentioned.

Figure 2: Please mention DI, AGF and AGS used in the figure axis.

Line 156: Degree symbols should be superscript than subscript. Besides, authors should justify the data present in Table 2 which correspond to 15 deg C, while the experiments were carried at 25 degC.

Number of figures should be reduced. It should be within 6-9 numbers.

Authors have described the experimental results. In most of the cases, the possible explanations have not properly been presented.

Authors should comment on the proposed mechanism. A schematic diagram on the proposed structure of the clay particles before and after adsorption may be presented.

Authors should comment on the morphology and the crystallinity of the clay particles upon the adsorption.

Reviewer 2 Report

In this study, the authors tried to study the chemistry of the clay mineral surface.  This study has no interest for the reader and is not important for the industry. They must do add more experimental work and present wider results. 

Reviewer 3 Report

  In this paper, the zeta potential measurement is used to gain an in-depth understanding of the surface charge development process of clay minerals in a natural reservoir environment. Meanwhile, the effects of fluid salinity, composition and oilfield operation on the development of the surface charge of clay minerals are introduced. 
  The paper is innovative and rich in content.My specific comments are as follows:
  1. "Error! Reference source not found." appears many times in the full text, please revise.
  2. In line 133, "with the mixture sonicated to ensure maximum interactions." needs to be supplemented with specific details of sonication, such as treatment time and number of treatments.
  3. In Figure 2, what are AGF and AGS? The symbols in the Figures should be consistent with the text description.
  4. In Figures 3-6 and 20-23, the ordinate of pH should be supplemented. Also, in Figure 7-11, the abscissa name is missing.
  5. In line 268, "This is attributed to the existence of an electrical double layer phenomenon." The explanation here is too brief, and it is recommended to expand the content.
  6. Lines 275-276, "More so, the electrical double layer effect can be observed in the case of SW..." How to observe?
  7. In Figure 7, under the FW and SW conditions, the zeta potential has a large mutation at 9 and 5, respectively. why?
  8. Where is the explanation about Figure 11?
  9. In Figure 12, why does the zeta potential under the action of Na ions decrease slightly from 25%Reduction to 50%Reduction?
  10. In section 3.3, it is recommended to supplement the specific formula and dosage of various salt solutions.
  11. In section 3.3, authors investigated the effect of different concentrations of cations and anions on zeta potential. But when the concentration of a certain cation is changed, the concentration of anion also changes. Will this not interfere with each other?
  12. In Section 3.5, why only chlorite and illite were selected for research? What about the results for kaolinite and montmorillonite?

Round 2

Reviewer 1 Report

Authors have tried to revise the manuscript in view of the reviewers comments. The revised manuscript may be accepted for publication.

Author Response

Thanks for your thorough review, the research design was reviewed

Reviewer 3 Report

The author has made corresponding changes, but the following improvements are still needed:
1. Figure 19-22 lacks the pH ordinate;
2. It is recommended to use double ordinates in Figures 3-6 and 19-22;
3. The answers to the original 7th, 9th, and 12th comments should be added to the corresponding positions in the paper.
